# Using Photovoice in a Mindfulness-Based Program to Understand the Experiences of Caregivers of Young Adults with Psychosis

**DOI:** 10.3390/ijerph192315461

**Published:** 2022-11-22

**Authors:** Herman Hay Ming Lo, Ken Ho Kan Liu, Wing Chung Ho, Elsa Ngar Sze Lau, Man Fai Poon, Cola Siu Lin Lo, Hillman Shiu Wah Tam

**Affiliations:** 1Faculty of Health and Social Sciences, Department of Applied Social Sciences, Hong Kong Polytechnic University, Hong Kong; 2Professional Practice and Assessment Centre, Department of Applied Social Sciences, Hong Kong Polytechnic University, Hong Kong; 3Department of Social and Behavioural Sciences, City University of Hong Kong, Hong Kong; 4Department of Education Administration and Policy, Chinese University of Hong Kong, Hong Kong; 5Baptist Oi Kwan Social Service, Hong Kong; 6Castle Peak Hospital, Hospital Authority, Hong Kong; 7Heartfelt Listening Counselling Space, Hong Kong

**Keywords:** family caregivers, photovoice, mindfulness-based intervention, psychoeducation, psychosis

## Abstract

Studies have consistently shown that family caregivers experience caregiver burden and depression when they provide care for family members with psychosis. Photovoice is a participatory action research method of fostering dialogues about personal experiences by sharing and discussing photographs that may improve our understanding about how a mindfulness-based family psychoeducation program (MBFPE) can reduce the caregiver burden and improve their caregiving experience. We explored the experiences of the participants in their use of photovoice in a MBFPE. We investigated whether the MBFPE program generated positive changes for caregivers of young adults with psychosis. Using photovoice, we collected qualitative data to help develop a unique contextual understanding of MBFPE program outcomes and generate novel ideas, insights, suggestions, and questions on the basis of participant’s perceptions. Participants in our pilot study and randomized controlled study of MBFPE were invited to participate in this photovoice activity. On the basis of a procedure developed for MBFPE, caregivers were invited to use photographs to express how mindfulness contributed to caregivers’ management of their caregiving stress and burden. Caregiver’s inquiry with the MBFPE instructors were transcribed for analysis together with the photographs. A grounded theory approach was adopted to analyze the photovoice images, participants’ reflections, and inquiries of photographs. Six themes were developed in understanding the lived experience of caregivers in participation of MBFPE: (1) I pay attention to the present moment; (2) I care about my family; (3) I trust my children; (4) I appreciate the connection with and support from nature and the universe; (5) I observe my worries and guilt and learn not to be reactive; and (6) I find space in offering care and exercising self-care. The application of photovoice can offer an additional approach to enhance the awareness and insights of participants in a mindfulness-based program. Specific guidelines may be developed to enhance the learning of participants.

## 1. Introduction

### Photovoice for Understanding Caregiving and Mindfulness Experiences

Studies have consistently shown that family caregivers experience caregiver burden and depression when they provide care for family members with mental health issues [1,2]. Psychosis, a serious mental illness, leads to significant impairments in social and family functioning of people in recovery [3], and its low remission and recovery rates often create a sense of hopelessness in the caregiving experience [4]. Many informal caregivers of psychosis within the family, who usually lack resources and mental health knowledge, experience helplessness and frustration at not being understood by others in their social circles [5,6]. The onset of psychosis is found as early as the age of 15 to 17, and the majority of individuals with psychosis reported their age as being between 20 and 30 [7,8]. 

An additional challenge for caregivers of young adults with psychosis is coping with stigma. Having a family member with psychosis often results in status loss and discrimination due to a process that begins with a labelling of human differences. Stereotyping or attributing of negative characteristics occur towards the persons in recovery who have been labelled with a socially salient difference [9]. Psychosis is stigmatized in many societies, including China, where its stigmatization is related to the traditional cultural emphasis on “face” [10]. Such face concern refers to a desire to preserve and maintain one’s social image and social worth, on the basis of one’s interpersonal context [11]. Chinese culture emphasizes the biological roots of mental illness and may consider an ill relative as a “bad seed” and a disgrace to their family [12,13], which may further exacerbate their stigma. As such, Chinese people are likely to keep their ill family member a secret in order to avoid face loss. The above psychological process is likely to create distress and subjective burden, symptoms of depression and anxiety [14], social isolation [15], and reduced community engagement [16]. Consequently, caregivers may become less motivated to seek help from mental health services, leaving their voices ignored and not heard. Given the marginalized status of caregivers, who often have few opportunities to share their views with others, research aiming to improve the understanding of caregivers of people with psychosis is needed.

Family psychoeducation (FPE) is a promising treatment option for dealing with family dysfunction, alleviating caregivers’ negative emotions, and managing psychosis relapses [17,18]. An FPE program usually imparts an understanding of the causes and symptoms of psychosis, cognitive behavioral techniques in stress management, and knowledge and practical skills in managing psychosis and family communication [19]. Family expressed emotion (FEE), which is defined as certain behaviors and attitudes exhibited by family caregivers toward their family member in recovery, namely, criticism, hostility, and over-involvement [20], has been linked to increased symptom severity and relapses of people with psychosis [21,22]. High FEE is more common in Asian contexts [23], probably due to Asian parents’ tendency for greater involvement in their adult children’s lives [24,25]. 

Inspired by the recovery model of psychosis, which is a response to earlier treatment models focusing on illness labeling, symptom control, and expert-centered treatment [26], mental health professionals are advancing an intervention approach that can cultivate resilience, hope, and empowerment [27,28]. Mindfulness, which is consistent with the holistic view of the recovery model, has been introduced to FPE [29]. Mindfulness-based interventions, which emphasize purposefully paying attention in the present moment non-judgmentally [30], have been found to reduce stress by promoting recovery and lessening negative self-referential processing [31]. There is also growing interest in applying mindfulness-based interventions to support parents by reducing their parenting stress and promoting positive parenting behaviors [32,33]. A mindfulness-based FPE (MBFPE) program acknowledges that recovery cannot focus only on the person in psychosis but also must include the well-being of caregivers and other family members. 

Meanwhile, there is growing attention in applying the qualitative and participatory research method of photovoice to give people, especially those in disadvantaged groups, the opportunity to record and reflect on their strengths and concerns. Photovoice is an innovative method of fostering dialogues about personal experiences by sharing and discussing photographs [34]. Photovoice has been applied in various health [35], family [36], and community research projects [37,38]. The purposes of photovoice include (1) facilitating community participation in advocacy, (2) enhancing awareness of community demands, and (3) boosting individuals’ sense of empowerment [39]. 

In an MBFPE program, caregivers’ photographs become a means to tap into their daily realities and the self-defined meanings and significance. As a participatory research method, photovoice allows participants to share their worlds through the medium of photography and through their own verbal responses to the images. The distinctive advantage of using photographs is the ability to capture lived experience. After participating in an MBFPE program, the participants learned to take photographs in a mindful and creative way so that the photo taking process could further enhance their awareness and insights [40]. The photovoice method can allow caregivers to make their voices heard regarding the recovery process, which involves caregivers, people in recovery, and the family. In a MBFPE program, caregivers have opportunities to mindfully reflect on their caregiving experiences and engage in dialogue with the mindfulness-based instructor and other participants to express their feelings about the photographs and perceptions about their situation. With the caregivers’ consent, their photographs and voices can be shared with others, including mental health professionals, as well as other people in the community. Smartphone cameras allow caregivers to capture meaningful moments in their lives and take the initiative to share their photographs and voices in an MBFPE program. On the basis of themes suggested by instructors, the caregivers contribute to the group discussion and provide mindfulness-based instructors and mental health professionals with a contextualized and equal environment in which to create or co-construct knowledge about how mindfulness can benefit their families [41]. 

In all mindfulness-based programs, instructors inquire into the participants’ experiences of practicing mindfulness during classes and at home [30,42]. This format aligns with the process of using photovoice in groupwork. With the participatory nature of photovoice, an MBFPE instructor can engage in inquiry with caregivers to enhance their awareness and understanding of the MBFPE program. As suggested by researchers and practitioners, photovoice can thus function as an educational tool for reflecting on how mindfulness brings changes to the daily caregiving experience of the participants [43]. The voices of caregivers can also inform community-service providers on how to improve the family and mental health practices of caregivers supporting family members with psychosis [44,45]. 

In this study, we explored the experiences of the MBFPE participants in their use of photovoice. The objective was to test whether the MBFPE program generated positive changes for caregivers of young adults with psychosis. We also collected qualitative data to help develop a unique contextual understanding of MBFPE program outcomes and generate novel ideas, insights, suggestions, and questions. Conducting qualitative analysis helped to ensure the internal validity of the intervention [46] as a supplement to our quantitative study of the program.

## 2. Materials and Methods

### 2.1. Study Design

A multi-site randomized controlled trial (RCT) with a mixed-methods design was carried out in 2019 to investigate the outcome of MBFPE among Chinese caregivers of young adults in recovery who experienced onset of psychosis in the past three years. Before successful application for large-scale research funding, we used a university internal seed funding to conduct a pilot study from 2017 to 2019. The effects of the MBFPE program were tested using a two-armed RCT comparing the MBFPE program with an ordinary FPE program. Both programs included a 1 h psychoeducation video, with an additional hour included in the MBFPE program for mindfulness exercises, inquiry, and photovoice (for more details of the study, see [29]). Instructors for MBFPE were practitioners with a qualification in social work, psychology, or family therapy, and they had completed professional training in mindfulness-based cognitive therapy. All instructors had experience in teaching mindfulness-based programs for at least two years. We included photovoice in the MBFPE program to explore how mindfulness contributed to caregivers’ management of their caregiving stress and burden. After transcribing the recording of photovoice activities in the MBFPE program, we analyzed the photos with participants’ description by using the grounded theory approach, an inductive method allowing us to discover new theory from observation [47]. We recruited 34 caregivers for the pilot study and 56 caregivers for the MBFPE in the RCT.

### 2.2. Photovoice Data Collection

This paper selected participants who joined an MBFPE from 2017 to 2022, including caregivers who joined the program in the pilot study and those who completed the RCT study. To encourage the MBFPE program participants to contribute candid and in-depth knowledge of the caregiving process and to explore the photographs unfolding, a series of procedures was developed as follows: In MBFPE sessions 2 to 5, exercises on the four photovoice themes (mindfulness of a pleasant moment, mindfulness of an unpleasant moment, mindfulness of a moment of difficult communication with the person in recovery, and the benefits of MBFPE) were given as homework assignments.The participants were encouraged to take photographs using their smartphones, according to guidelines offered at the end of these sessions.The participants were invited to provide titles and additional reflections on the photographs and share them in subsequent sessions. To facilitate group sharing, the caregivers were asked to send their photographs and reflections to the research team before the session. A PowerPoint slideshow was then prepared in sessions 3 to 6 to enable all of the participants to see the photographs clearly.Ten to fifteen minutes were allocated for inquiry into the photographs and the caregivers’ reflections in sessions 3 to 6. With the participants’ consent, the photographs, reflections, and content of the in-session inquiries were used for data analyses.

### 2.3. Photovoice Data Analysis

The author and the research team applied grounded theory to analyze the photovoice images, participants’ reflections, and MBFPE transcripts [47]. Conceptual categories were aroused through the data interpretation, and the research team maintained a reflexive mode about the prior interpretive frames, interests and research context, and relationships with participants in generating and recording empirical materials in the process of analyses [48]. The team watched the pictures and the videotapes of the MBFPE sessions and studied the transcripts of themes, categories, and concepts generated during the photovoice inquiries. Using a constant comparative method, the researcher refined the framework through categorizing, coding, and delineating categories and connecting them. The cycle of comparison and reflection on “old” and “new” material was repeated several times [49].

Through the analyses of photovoice that were contributed by caregivers, the research team members produced ideas of what and how mindfulness could be useful to caregivers to more community stakeholders and share responsibility for the advancement of knowledge. The research team shared these reflections and analyses with the MBFPE instructors and invited them to clarify, elaborate upon, and critique the interpretations in individual meetings. Knowledge is co-constructed and improved by an open, collaborative workspace, and is democratized among caregivers, MBFPE instructors, mental health professionals, and researchers [50]. Finally, an additional meeting was arranged for more mindfulness-based instructors who were clinical practitioners and not involved in this study. A sharing session on the findings of photovoice was arranged. Feedback and comments were collected to strengthen the reliability of the qualitative study analyses. Collaboration with stakeholders and the democratization of knowledge construction are strategies for enhancing research credibility for participatory action research [51].

## 3. Results

### 3.1. Profile of the Caregivers Involved in Photovoice in This Study

The photographs selected for this study were drawn from those submitted to the MBFPE program instructors by 13 caregivers. During the pilot study and RCT, about one-third of the caregivers submitted photographs to the instructors and the research team. Some of these were excluded from further analysis due to incomplete data or inadequate clarity in the photographs or their reflections. Table 1 gives a summary of the profile of the photovoice participants. All caregivers were parents, including 10 mothers and 3 fathers. All the people in recovery were young adults aged between 15 and 30. When asked to indicate the diagnoses of their family members in recovery, some caregivers referred to “thought and perceptual disorder”, which is a common label in Chinese for describing a serious mental disorder; for simplicity, we used “psychosis” as the diagnosis in these cases. In this study, both “thought and perceptual disorder” and “psychosis” referred to schizophrenia spectrum, bipolar, and other related psychotic disorders listed in the Diagnostic and Statistical Manual of Mental Disorders, 5th Ed. (DSM-5, [52]).

### 3.2. Voices of the Caregivers

We identified six different themes among the voices capturing the experience of the caregiving and recovery process. To illustrate the themes for each of these voices, we selected representative photographs with the participants’ descriptions. All of the photographs shown here were shared in the MBFPE sessions. Pseudonyms are used here to maintain the participants’ anonymity.

#### 3.2.1. Voice 1: I Pay Attention to the Present Moment

The caregivers were mindful of the positive and negative events in their lives. Although they faced challenges, mindfulness practice allowed them to focus on and stay open to the present moment. Paying attention to the present moment opened space for them to take better care of themselves. Many caregivers also expressed the importance of seizing the moment and acknowledging the positive experiences of caregiving.

Reflecting on the photograph in Figure 1, Participant A recalled that the weather was fine while she wandered along the seaside. At that moment, she could see the positive side of things: 


*“At dawn, it was cloudy. I couldn’t see the sunrise, but it was not so hot. I comforted myself and thought it was okay.”*
(Participant A)

Reflecting on the photograph in Figure 2, Participant B said that she enjoyed preparing meals with her son, especially when he was satisfied with the dishes: 


*“In the past two or three years, I have needed to explore different dishes for him and I felt happy and satisfied when he pigged out on all the dishes.”*


It is interesting to notice that caregivers shared many photographs about pleasant activities relating to preparation or sharing of food with their family member. Eating is also one of the important informal practices in mindfulness training, and it offered much awareness and insights to caregivers: 


*“In the morning, we had breakfast after testing for COVID-19. We were quite unhappy about the testing, but we enjoyed the nice breakfast and sunshine. After that, we took a walk and went shopping in the new recently launched store near our place.”*
(Participant C)


*“I went to Discovery Bay to shop for food. I felt very happy when I found some attractive items. Like yesterday, there were fruit, bread, tomato soup and a watermelon on the dining table when I got home. All this food made me feel delighted.”*
(Participant D)

#### 3.2.2. Voice 2: I Care about My Family 

The caregivers saw their care for the persons in recovery as one of the core elements of their recovery. They expressed the intention to cultivate a caring family and spend more time caring for family members. All of the family members cherished the moments of being supported and accompanied.

Participant E used the condition of a ship (see Figure 3) as a metaphor for the challenges that people face and the importance of being mindful:


*“I can only work without disturbance at night. I saw a ship with the word “dream” on the side, and I thought it was meaningful. Sometimes, having some private time is good for self-reflection. I use this ship as a metaphor to describe our lives. The ship is stable when there is no heavy wind; otherwise, the situation will become challenging. There are many conflicts within the family, but it is important to be mindful of every moment. I hope that I will continue believing in my dream: may my family be peaceful.”*
(Participant E)

Participant E used the house in the photograph in Figure 4 to illustrate what an ideal family looks like: 


*“I hoped that my family would become more harmonious, peaceful, caring, and inclusive. This house looked old on the surface, but I thought that the important things were the characteristics I mentioned. Although I am not achieving these things at this moment, I keep learning, including how to accept myself.”*
(Participant E)


*“The memorable reward of this course was that I learned how to live in the present moment. In our life, there are different happy and challenging moments. The important thing is to take care of myself so that I can support my family.”*
(Participant F)

#### 3.2.3. Voice 3: I Trust My Children

The caregivers expressed trust in their children’s abilities, despite their health conditions. Compassion and hope were protective factors for the caregivers’ mental health and helped them to devote themselves to giving care. The voices of the caregivers also expressed the desire for the persons in recovery to grow through self-care. As mindful caregivers, they regarded it as important to support their children in becoming independent and willing to try doing things on their own. 

Reflecting on the photograph in Figure 5, Participant F expressed relief that her son was doing well and was inspired by reading the word “hopeful” in his homework. 


*“I observed my son doing a homework assignment for a design course. … I thought the word “hopeful” was meaningful and I wanted to share it because I feel like people do not smile often. I feel hopeful when I know that my son is studying hard and has found something he likes. As a family member, I have learned how to accept my son’s mental problem and to get along with him. Hope is always there.”*
(Participant F)

Participant G was a caring father who believed that his son could be as brave as the paraglider in the photograph in Figure 6, despite the challenges of the recovery process:


*“I hope my son can become a ‘warrior’ like other people who have recovered, and I think positively. My son liked this picture on Facebook; I thought it was a good sign that he appreciates others’ recovery stories. I think that the person in recovery in this picture is the embodiment of courage because they have overcome many difficulties. In terms of recovery, I know that my son will not become ‘normal’ again but I hope that he will at least maintain a stable condition.”*
(Participant G)

Participant H also emphasized the personal growth of her son. She took a photograph of a construction site on a cloudy day and referred to it when sharing the struggles with her son’s recovery since he was diagnosed with autism at a young age:


*“I wanted to capture this cloudy day. My son has had ADHD and autism since he was a kid; his ADHD meds and the traumatic experience in adolescence caused his schizophrenia, which was validated by medical evidence. However, I believe that the worst experiences can be our best opportunities for growth. …. He knows how to cook now, and I am thankful that he has undergone personal growth and is willing to take his medicine. He no longer feels tired after taking his medicine. I am worried about the upcoming 2022 public exam, but I believe my son will achieve good academic results.”*
(Participant H)

#### 3.2.4. Voice 4: I Appreciate the Connection with and Support from Nature and the Universe

The caregivers were thankful to nature and the universe for helping them learn to be mindful. After practicing mindfulness, the caregivers became more aware of the support they received from friends, nature, and their beliefs. The caregivers expressed that apart from family, nature and the universe provided valuable support that increased their resilience in the process of recovery.

Participant I took a picture of a snail during her morning walk (see Figure 7) because it reminded her of the progress of her son’s recovery. It reminded her that one can learn from nature and one’s surroundings: 


*“I am used to waking up at 5 a.m. and then I will take a walk. I am grateful and thankful to nature and my son who showed me the importance of being flexible and resilient.”*
(Participant 1)

In the photograph in Figure 8, the Chinese words on the cross mean “faith”, “hope”, and “love”. Participant F shared that her religious beliefs gave her the power to overcome many obstacles:


*“After my son suffered from some health problems, I started developing religious beliefs. Subsequently, my sister and my mother developed religious beliefs, too. It helped her to overcome many difficulties in life.”*
(Participant F)

#### 3.2.5. Voice 5: I Observe My Worries and Guilt and Learn Not to Be Reactive

The caregivers were worried about their family members in recovery, but with mindfulness practice they learned not to respond reactively. They remained open and receptive to ways of supporting the recovery of the people they were caring for, although they were uncertain about what they could do and what would be helpful. They learned to observe them and ask useful questions to gain more understanding of them. Aware of the possible negative consequences of being reactive, they believed it was important to offer some space and find more comfortable ways to get along together.

The photograph by Participant J, shown in Figure 9, reflected her desire not to disturb her son even, although she was worried about his looking at the screen for a long time. She had learned not to overreact:


*“I wanted to talk with him more and to understand him. … I was uncertain whether he was in class or doing homework. He looked happy but it is not good to spend a long time on the phone.”*
(Participant J)

Participant K took a photograph of a book about parenting a defiant child that his wife had given him before she died of cancer (Figure 10). He shared that his wife’s words were a constant reminder that he needed to give his son time and space for growth and recovery. Although he had some difficulties communicating with his son, he did not give up:


*“This was the book that my wife had left, reminding me that I had not taken enough responsibility of taking care of my son. It was hard for me to compensate for what I had missed, and I found it difficult to communicate with my son. I wanted to encourage my son to do something, but it was always in vain. Despite the failures, I kept trying; for example, I encouraged him to do more exercises with a friendly tone as I believed that doing these exercises would be beneficial for him. Conversations with my son always became a conflict and we ended up silent. Therefore, I needed to prepare myself before talking to him as I had an inner struggle. I had to control myself. When he lost his temper, I chose to keep quiet, avoiding intense quarrels if possible. I hope that what I taught him has planted a seed and that he will eventually understand.”*
(Participant K)

Reflecting on his photograph of a pill box (Figure 11), Participant L expressed his concerns about whether his daughter would take her medicine on time and take care of herself. He had learned how to remain calm, despite being worried:


*“My daughter moved to a dorm last week. I was worried about whether she would take her medicine on time. If she forgets, it will increase the chance of relapse. I contact her every day to find out how she is. One morning, I called her and she said that she had taken the medicine. However, that night, I noticed that she had not taken it. I was worried and I told her to take the medicine immediately. Then I reminded her to take her medicine next time and talked about the negative consequences of not taking her medicine on time. I was a bit annoyed because she did not tell me the facts. Luckily, the medicine was kept in a box so I could monitor it. I felt anxious about whether she could take care of herself and about the possibility of relapse. I have learned not to overreact in front of my daughter, even while feeling worried.”*
(Participant L)

Participant J also took a picture of his son’s pill box and described it as “frustrating and worrying to see him taking medicine every day”.

Participant M photographed his son’s profile picture on an instant messaging application, which was based on a science fiction character: “It is the profile picture of my son. I felt so uncomfortable while looking at this picture and believed that there was some hidden meaning in his profile picture”.

#### 3.2.6. Voice 6: I Find Space in Offering Care and Exercising Self-Care

The caregivers observed conflicts between their children and within the family but tried to hold their strong emotions within themselves. They recognized the needs of the people in recovery and understood the importance of calming down. In the midst of challenges and stressful moments, they realized their vulnerabilities and need for self-care.

Participant E described a misunderstanding with her daughter over the pot in her photograph (Figure 12). She believes that her son needs more space and care for his recovery, even though other family members may not understand:

*“When I came home, I saw that the pot was open, and I asked my daughter why. My daughter got angry and complained that I did not understand her. … As he [my son] was sick, my daughter thought I cared for him only, and this sometimes made her angry. … I felt that I paid more attention to my son. This made my daughter think I was biased. … We all maybe need to understand and tolerate each other more. I am not biased at all. This scenario may happen time and time again.”* (Participant E). (Authors note: in Chinese, an exploded pot has often been used as a metaphor to describe a heated quarrel or fight.)

Participant B felt thankful to have made the bowl of congee in her photograph (Figure 13), which was made from ingredients provided by her relatives. She also learned to respect her daughter’s preference:


*“It was a bowl of congee. I received some pumpkins from my relatives in mainland China. I felt grateful that they were healthy. Actually, I cooked this for my daughter. However, she did not like it and she even complained about it in the interview with the clinical psychologist. I was fine and I realized that there were different expectations between my daughter and myself.”*
(Participant B)

Participant B also took pictures of a program book for a class that she was interested in and the online enrollment form for a marathon. She was somewhat upset that she missed the class and the enrollment deadline for the marathon, but she gently reflected on this: “I became more forgetful as I missed a class and the deadline of registering for the marathon. I started criticizing myself. I reflected that I relied too much on the phone.”.

### 3.3. Feedback and Comments from MBFPE Program Instructors

Three instructors were involved in delivering MBFPE to caregivers in this study. All of them received full foundational training in mindfulness-based cognitive therapy. After the first author presented the data analysis of photovoice to them, they were invited to share their feedback and comments about the analysis and their experiences in applying photovoice in an MBFPE.

Instructor A was a supervisor of an integrated community mental health center and an experienced mindfulness-based instructor. He found the photovoice offered a unique, non-verbal channel for caregivers to express their experiences, instead of relying on the inquiry of practice experience in an ordinary mindfulness-based program. The activity provided an opportunity for participants to review their pleasant moments, unpleasant moments, and other experiences in caregiving. Through the process of taking photographs and sharing in the group session, caregivers have a dialogue within themselves through taking a pause and a few breaths and enjoying a space in paying attention to their own experiences in caregiving. Photography is a good means for exploring personal meaning and facilitating social support that may be used to promote growth and resilience of caregivers.

Instructor B was a clinical psychologist at a psychiatric hospital. She expressed that she felt touched when she received the photographs from the participants, as she could feel their lived experiences and their struggles with their children, which were beyond words. She could feel their sorrows, regrets, and bitterness through their perceptions and how these could be embraced and transformed in the process of mindfulness training and with the compassionate support of the participants. The photovoice exercise was a new experience to her as an instructor and it spoke the experience of the participants in a way that is close to their hearts. However, she also expressed a limitation that she had no experience in delivering a photovoice exercise in a class, and she was not certain whether she had delivered the activity appropriately.

Instructor C was a certified family therapist working at an NGO and a former artist who had received extensive training in photography. He found that photography is a suitable means to integrate mindfulness training in a psychoeducation program. However, he also expressed a concern that the course structure needs to be modified to allow more time and space for caregiver’s voices, and he felt a time pressure in processing the experiencing in photo taking.

### 3.4. Feedback and Comments from Other Mindfulness-Based Program Instructors

The first author gave a presentation at regular meeting for mindfulness-based program instructors in August 2022. Thirteen instructors attended the meeting, all of whom had completed basic professional training in mindfulness-based cognitive therapy or mindfulness-based stress reduction programs.

The 13 instructors showed their appreciation for the use of photovoice in a mindfulness-based program. They believed it could serve as a useful tool for caregivers, especially those who are less articulate in sharing their feelings and thoughts through language in order to communicate their caregiving experiences. They considered the photographs as a useful activity for a mindfulness-based program instructor to facilitate the emergence of awareness and insights.

One instructor shared his thinking on suitability to deliver an inquiry in photovoice in a MBFPE session. For an art therapy practitioner, the conversation would focus on the unfolding of underlying meaning and meanings and concerns. As a mindfulness-based instructor, one may encounter challenges between staying in the present and thus align themselves with the principle of mindfulness, instead of using clinical skills in exploring participant’s past experiences, as in many art therapy sessions.

## 4. Discussion

Family interventions for psychosis have focused on adjusting expressed emotion [21,22]. In this study, we found that the caregivers appreciated learning how to use mindfulness to reduce their criticism, hostility, and over-involvement by regulating their strong emotions. Such learning is supported by voice 1 (paying attention to present moment), voice 4 (appreciating the connection and support of nature and universe), voice 5 (observing strong emotions and learning to be reactive), and voice 6 (exercising self-care). Although some caregivers acknowledged the difficult experiences in caregiving, they learned to hold their intense emotions with the support of mindfulness. From the MBFPE, their commitment to family care was heard (theme 2), and they learned to trust their children (theme 3). All six themes validate the positive outcome of the program. Although photovoice does not involve raising concrete questions about whether caregivers exhibit highly expressed emotion and how they handle this, the photographs and their reflections on them demonstrated their ability to manage difficult emotions and become more aware of the impact of these emotions on the person in recovery.

A possible benefit of MBFPE, and psychoeducation in general, is to promote positive caregiving experiences. The photovoice theme of mindfulness of a pleasant moment reminded the caregivers in particular of the positive experiences amidst the struggles and challenges of caregiving and recovery. The qualities of mindfulness and mindful parenting—acceptance and trust—also reminded the caregivers that they and the person in recovery will experience growth and develop resilience [30,53]. Positive caregiving experiences have been shown to be positively associated with caregivers’ quality of life [54]. The value of MBFPE and photovoice in reminding caregivers how much they care about their family might also help to reduce the self-stigma, which was often experienced when caregivers endorsed the external stereotypes about people in recovery and caregivers, as well as in a pathway for recovery and better quality of life for the whole family.

The idea of using photographs and photovoice to promote mindfulness has been raised by a few researchers and practitioners [55]. The promotion of embodiment in mindfulness and the power of images in emotional processing and transformation are two useful resources in therapeutic work. However, the first author and a mindfulness-based instructor from whom feedback was obtained share a concern over whether the use of images that elicit strong emotional memories might overshadow the emphasis on embodiment and present experience in mindfulness. Photovoice was not originally developed as a tool for psychotherapy or therapeutic work, and instructors should not spend too much time analyzing the meanings of the images. To maximize the benefits of using photovoice, experienced practitioners should develop sample questions and guiding principles for the future use of photographs in alignment with the best practice in mindfulness-based programs.

Photovoice is suited for use as an informal practice, such as those of mindful eating or mindfulness of one’s daily routine, in mindfulness-based programs. It can serve as a bridge to carry the awareness learned from mindfulness training into daily lives. Assigning photovoice tasks may also remind participants in a mindfulness-based program to pause and relax, to be aware of the present moment, and to recall what they learn when they take photographs. Applying photovoice in a psychoeducation program such as MBFPE should not diverge from the original idea of the method, which focuses more on empowerment. Nonetheless, recent studies have demonstrated that the integration of mindfulness with photography can promote creative learning and health promotion [56,57]. On the basis of these findings and the findings presented in the present study, further promotion of the use of photography or photovoice in mindfulness-based programs is recommended.

Lastly, we highly appreciate the use of photovoice as a useful tool to collect qualitative data about the MBFPE outcome, on the basis of the participant’s perception. In the protocol of an outcome study of MBFPE, we embedded photovoice in a RCT that has included many standardized measures selected by the researchers. Photovoice offers a distinctive role in that we can understand how caregivers benefit from mindfulness exercise from a participatory action research method. We find it can be developed as an organic part of any other mindfulness-based programs, and participants can share what they experience and learn in a photo-taking activity that can be closely integrated into their daily lives.

However, the positive evaluation of the application of photovoice in this study should be interpreted cautiously due to certain limitations, of which three are identified here. First, during the implementation of the RCT, there were five waves of COVID-19 in Hong Kong. As many caregivers showed interest in the trial but were reluctant to attend a face-to-face program, the sample size was less than half of what was expected [29]. As the use of a small sample may lead to bias, the application of photovoice should be evaluated with larger samples in future studies. Second, in the study protocol of the RCT, we recruited caregivers of young adults with psychosis who had reported onset in the last three years. The aim of setting these inclusion criteria was to test the effects of mindfulness in supporting families who had recently experienced the crisis of illness onset and were making a transition to recovery. However, we realized that the recent onset group often encountered high stress, as many young adults in recovery were experiencing an unstable mental state and frequent relapses. Therefore, some caregivers meeting these criteria would find it very challenging to commit to a psychoeducation program and practice mindfulness. Further studies should include a broader range of caregivers and consider the inclusion of caregivers of family members with chronic mental health issues, such as chronic depression or dementia. Third, in the original study design, the participants were to be gathered for a nine-month follow-up session, in which instructors would present the photographs to the caregivers again and facilitate sharing and learning from their experiences after the MBFPE program. However, these face-to-face meetings were cancelled because of the COVID-19 pandemic, and no further qualitative data relating to photovoice was collected. The long-term effects of including photovoice in an MBFPE program therefore remain unclear, and it is uncertain as to whether the awareness and insights gained from photovoice can be sustained.

## 5. Conclusions

In this study, photovoice documented the perception of benefits of a mindfulness-based psychoeducation program for caregivers of young adults with psychosis. Photography has been widely used in education and health promotion in view of its convenience and the creativity that is involved. Such a method should be further promoted among researchers and practitioners, and its potential values in integrating with a mindfulness-based program should be further investigated.

## Figures and Tables

**Figure 1 ijerph-19-15461-f001:**
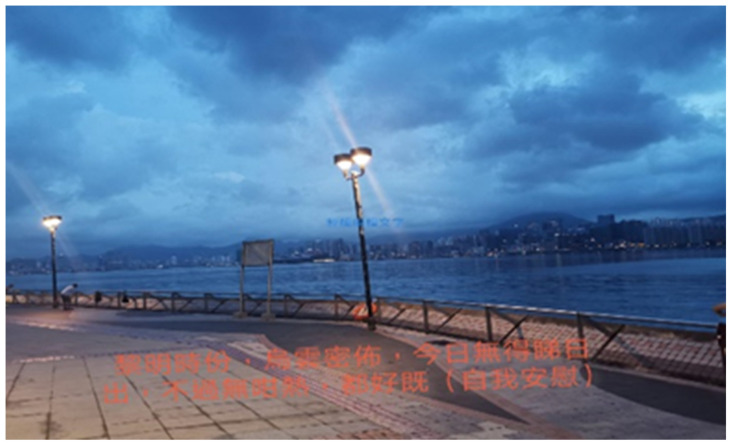
Photograph by Participant A.

**Figure 2 ijerph-19-15461-f002:**
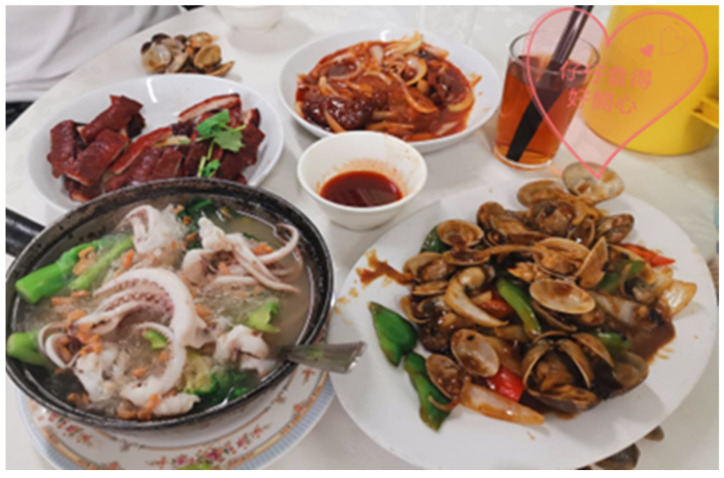
Photograph by Participant B.

**Figure 3 ijerph-19-15461-f003:**
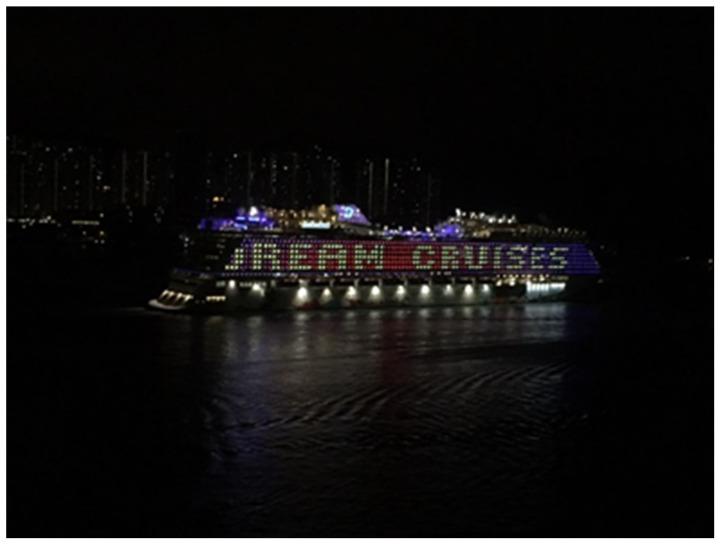
Photograph by Participant E.

**Figure 4 ijerph-19-15461-f004:**
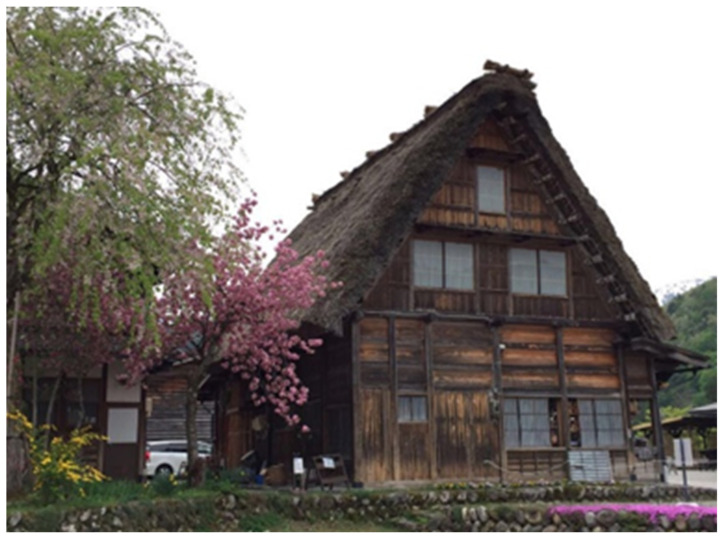
Photograph by Participant E.

**Figure 5 ijerph-19-15461-f005:**
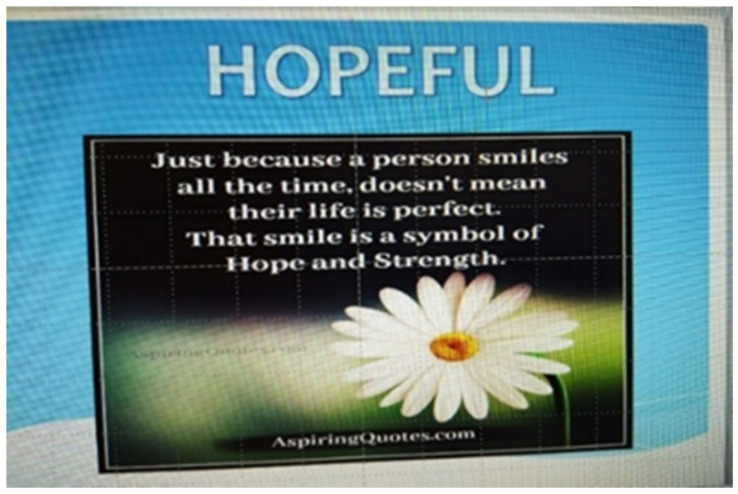
Photograph by Participant F.

**Figure 6 ijerph-19-15461-f006:**
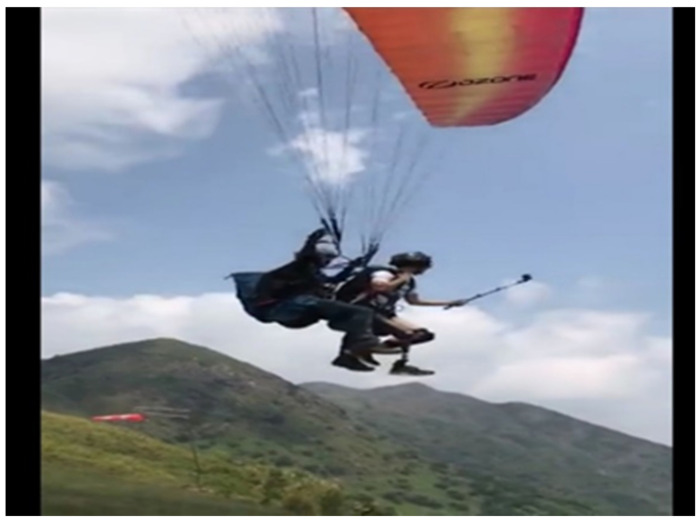
Photograph by Participant G.

**Figure 7 ijerph-19-15461-f007:**
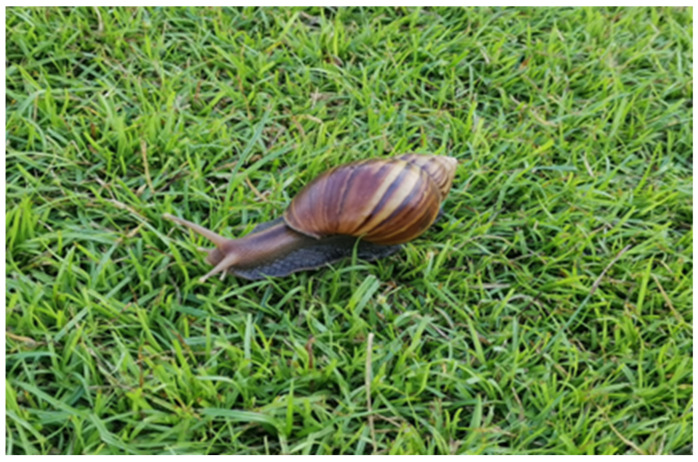
Photograph by Participant I.

**Figure 8 ijerph-19-15461-f008:**
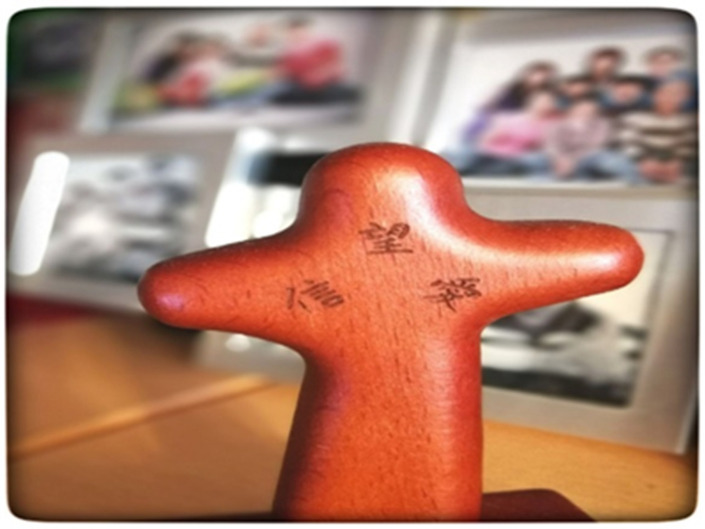
Photograph by Participant F. The words on the wooden cross is faith, hope, and love.

**Figure 9 ijerph-19-15461-f009:**
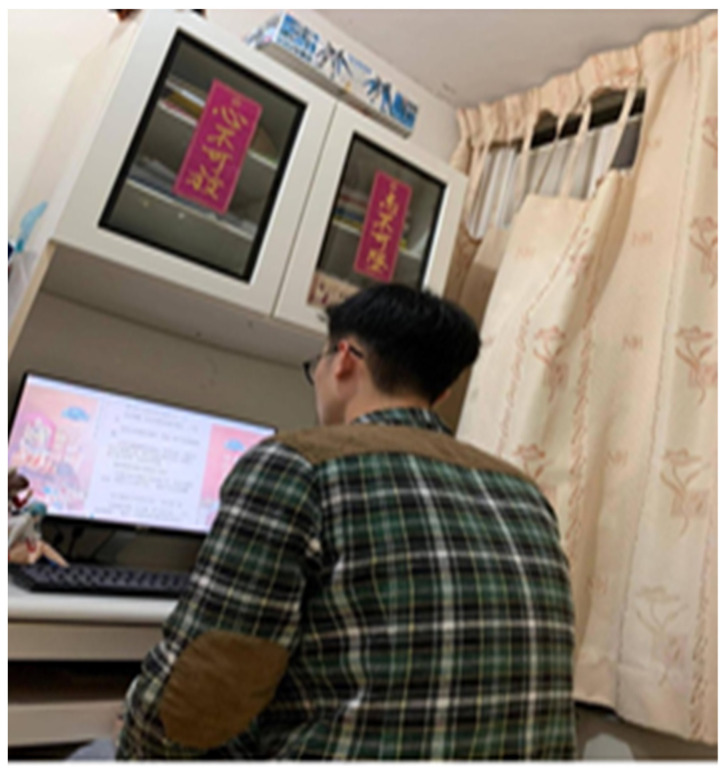
Photograph by Participant J. The Fai Chun (traditional decoration) on the bookshelf are “High but not falling, Heart but not losing”.

**Figure 10 ijerph-19-15461-f010:**
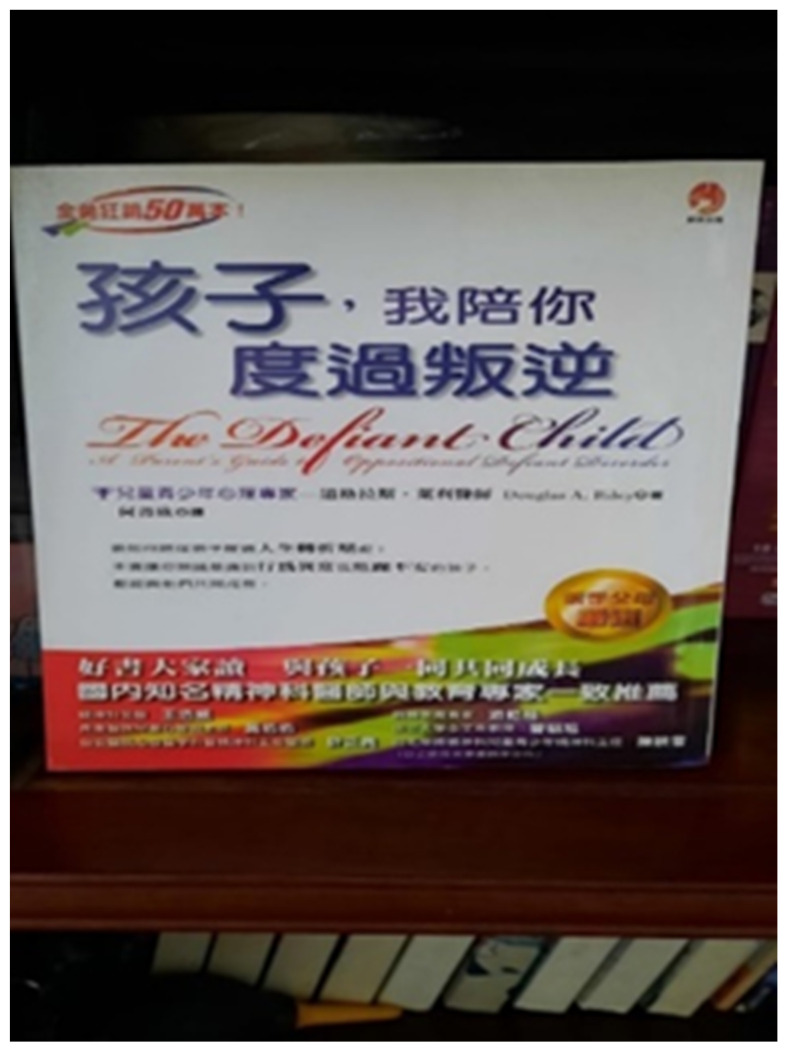
Photograph by Participant K. This is the book cover of the Chinese version of “The Defiant Child” and the subtitle in Chinese is “Child, accompany you through the period of rebellion”.

**Figure 11 ijerph-19-15461-f011:**
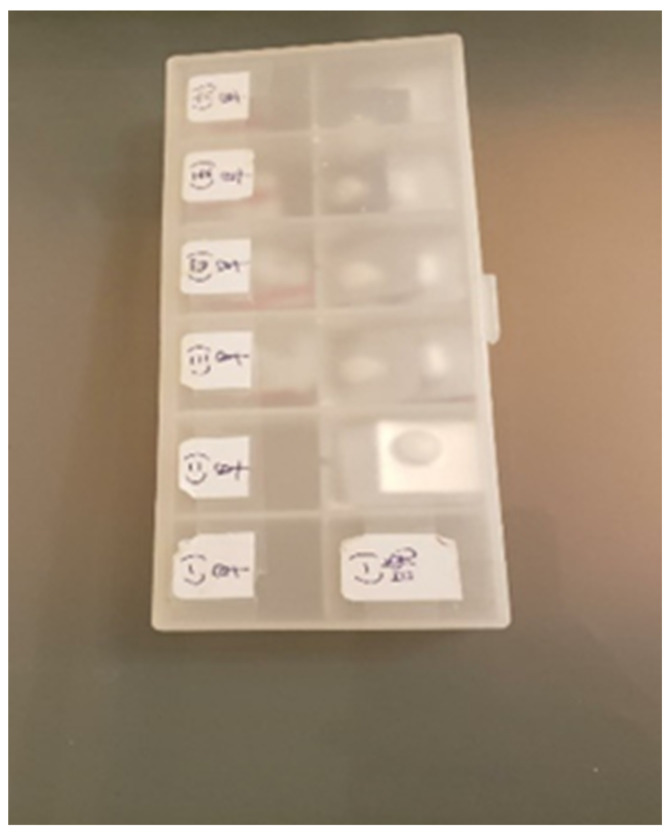
Photograph by Participant L. The marks on the pill box are Monday to Saturday, morning and night.

**Figure 12 ijerph-19-15461-f012:**
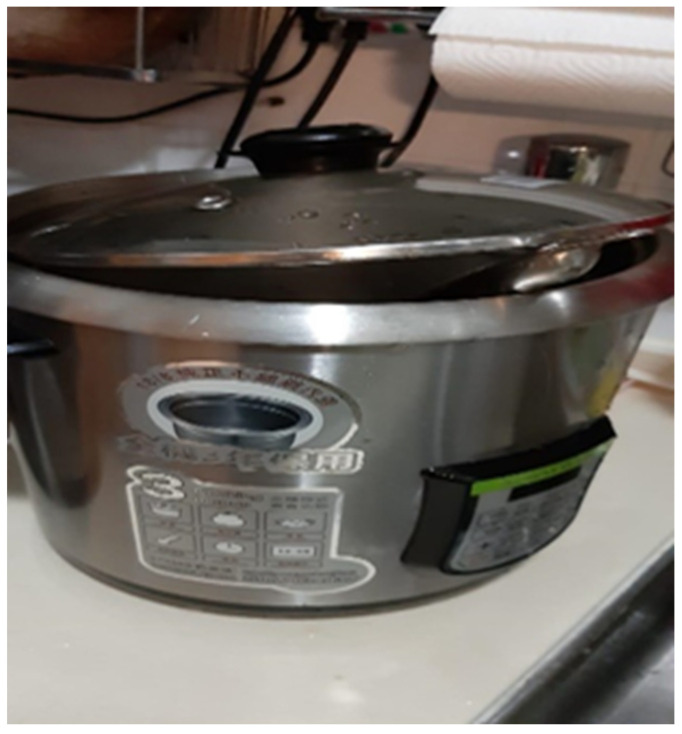
Photograph by Participant E.

**Figure 13 ijerph-19-15461-f013:**
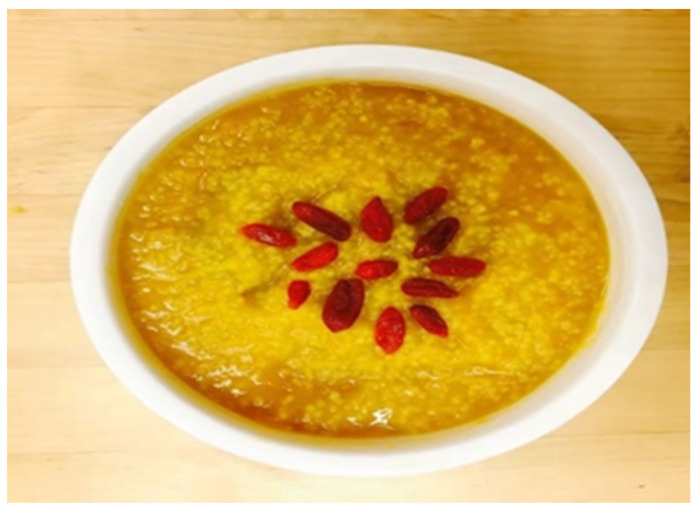
Photograph by Participant B.

**Table 1 ijerph-19-15461-t001:** Profile of the participants who submitted photographs and were selected for data analysis.

Participant	Sex	Age	Relationship to Person in Recovery	Age of Person in Recovery	Diagnosis of Person in Recovery
A	F	41	son	15	Schizophrenia
B	F	45	son	18	Schizophrenia
C	F	49	daughter	22	Schizophrenia
D	F	60	son	30	Schizophrenia
E	F	53	son	27	Psychosis
F	F	60	son	26	Psychosis
G	M	57	son	27	Psychosis
H	F	48	son	21	Schizophrenia, ASD, ADHD
I	F	52	son	24	Schizophrenia
J	F	53	son	19	Schizophrenia
K	M	65	son	27	Schizophrenia
L	M	55	daughter	21	Psychosis
M	F	46	son	19	Schizophrenia

## Data Availability

Not applicable.

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
