# Peer review of "Using Photovoice in a Mindfulness-Based Program to Understand the Experiences of Caregivers of Young Adults with Psychosis"

_ijerph, 2022, doi:10.3390/ijerph192315461_

Round 1
Reviewer 1 Report
To authors
In this study, authors used a new technique of Photovoice combined with mindfulness-based program to caregiver’s educational program. I think this trial would expand the effect of mindfulness training and care for suffering caregivers, because discussions using Photovoice would be expected as peer counseling or art therapy that might be able to engage deeper consciousness of participants. This research is worthy of recognition, and I look forward to its further development.
However, I think the following changes are necessary.
Major comments:
P.3, L.137 2 Materials and Methods
Although you described that “a grounded theory approach was adapted” in this study at abstract, you didn’t mention it in the description of “methods”. Would you mention how the analyst in this study analyze the participants' statements in order to demonstrate the objectivity as the scientific study?
And inclusion criteria you mentioned in discussion, such as “we recruited caregivers of young adults with psychosis who had reported onset in the last three years”, would be also better to be shown in the section of methods.
P.3, L.141
In academical research, I think it might be better to clarify what “psychosis” is. Because "psychosis" includes diseases with a good prognosis that is different from schizophrenia. You mentioned that you use word “psychosis” according to participant’s statements in P.4, L.182. I wondered if there was a difference in the realities of the person the participants cared for between in the case of 'schizophrenia' and in the case of 'psychosis'. Do you think the prognosis of the person the participants cared for is not affect the results of this study?
Minor comments:
p.1, L19
Wouldn't you add the word “family” in the part of “mindfulness-based psychoeducation program ” in order to match the abbreviation, MBFPE?
P.2, ,53
Is the word “rresults” misspelling?
p.3; L109, P.3; L147, L159, P.15;L.519, L522,
Is the abbreviation, MBPFE same meaning as MBFPE?
If so, could you unify the abbreviations into one?
Otherwise please explain the difference between MBPFE and MBFPE.
P.3, L.104
It is difficult for me to understand the subject in the grammatical sense at the part “ and for participants who took photographs in a mindful and creative way were more appreciative and motivated would experience a better mood than those who did so in a neutral manner”.
Could you make clear expression?
P.5, L.206
Is “Figure 3” “Figure 2”?
P.6, L.227
Is ”Figure 4” “Figure 3”?
P.8. L.271
What is the meaning of @ in this sentence?
Author Response
Point-to-point responses to the comments of reviewers
Comments and Suggestions for Authors
In this study, authors used a new technique of Photovoice combined with mindfulness-based program to caregiver’s educational program. I think this trial would expand the effect of mindfulness training and care for suffering caregivers, because discussions using Photovoice would be expected as peer counseling or art therapy that might be able to engage deeper consciousness of participants. This research is worthy of recognition, and I look forward to its further development.
However, I think the following changes are necessary.
Major comments:
P.3, L.137 2 Materials and Methods
Although you described that “a grounded theory approach was adapted” in this study at abstract, you didn’t mention it in the description of “methods”. Would you mention how the analyst in this study analyze the participants' statements in order to demonstrate the objectivity as the scientific study?
And inclusion criteria you mentioned in discussion, such as “we recruited caregivers of young adults with psychosis who had reported onset in the last three years”, would be also better to be shown in the section of methods.
Responses:
Thank you for the useful comments. We have revised the manuscript in:
- We have expanded in the description of grounded theory approach in 2.1 and added 2.3 data analysis for explaining how can conduct the data collection and analysis using photovoice.
- we have added the inclusion criteria of the caregivers in 2.1 for clarity.
P.3, L.141
In academical research, I think it might be better to clarify what “psychosis” is. Because "psychosis" includes diseases with a good prognosis that is different from schizophrenia. You mentioned that you use word “psychosis” according to participant’s statements in P.4, L.182. I wondered if there was a difference in the realities of the person the participants cared for between in the case of 'schizophrenia' and in the case of 'psychosis'. Do you think the prognosis of the person the participants cared for is not affect the results of this study?
Response:
Thank you for your comments. In the original caregiver’s statements, some participants filled “thought and perceptual disorder” in the psychiatric diagnosis. It is a label created by psychiatrists in Hong Kong and is commonly used in the local community for describing a serious mental disorder. For simplicity we used “psychosis” in the manuscript. In this study, both “thought and perceptual disorder” and “psychosis” referred to Schizophrenia Spectrum, bipolar, and other related psychotic disorders listed in Diagnostic and Statistical Manual of Mental Disorders, 5th Ed. We do not have adequate data to answer the question about whether there is a difference in the realities of the person that participants cared for between the case of schizophrenia and in the case of psychosis/thought and perceptual disorder. Our general impression is that participants in this study involves those of individuals with psychosis with different prognosis and we have no evidence that there is a link between prognosis and their uses of photovoice.
We have provided additional explanation in line 214-217. It is expected to improve the clarity.
Minor comments:
p.1, L19
Wouldn't you add the word “family” in the part of “mindfulness-based psychoeducation program ” in order to match the abbreviation, MBFPE?
Response:
Thank you for your comment. We have double checked and corrected the full name and abbreviations across the manuscript. The word "family" is added and "MBFPE" is used across in the revised manuscript.
P.2, ,53
Is the word “rresults” misspelling?
Response:
The typo is corrected.
p.3; L109, P.3; L147, L159, P.15;L.519, L522,
Is the abbreviation, MBPFE same meaning as MBFPE?
If so, could you unify the abbreviations into one?
Otherwise please explain the difference between MBPFE and MBFPE.
Response:
Sorry for the mindless typo. All abbreviations have been corrected.
P.3, L.104
It is difficult for me to understand the subject in the grammatical sense at the part “ and for participants who took photographs in a mindful and creative way were more appreciative and motivated would experience a better mood than those who did so in a neutral manner”.
Could you make clear expression?
Response:
Thank you for your comment. The sentence is rewritten as follows:
After participating in a MBFPE program, the participants learned to take photographs in a mindful and creative way so that the photo taking process could further enhance their awareness and insights. (L107-109)
P.5, L.206 Is “Figure 3” “Figure 2”?
P.6, L.227 Is ”Figure 4” “Figure 3”?
Response:
Thank you for pointing out the typo. They have been corrected.
P.8. L.271 What is the meaning of @ in this sentence?
Response:
It is a typo and @ has been deleted.
Reviewer 2 Report
Thank you for your study and the opportunity to review it. I enjoyed it very much. I think it could be improved by offering more detail about photovoice as well as more specifics regarding the MBI used here (MBFPE), including the trainer/teacher's credentials/qualifications to teach the MBI.
These are my suggestions for strengthening your submission.
Author Response
Thank you for your comments.
We have expanded the explanation of photovoice and the data analysis of photovoice in 2.1 and 2.3.
Instructors for MBFPE were practitioners with a qualification in social work, psychology, or family therapy, and they completed a professional training in mindfulness-based cognitive therapy. All instructors had experiences in teaching mindfulness-based programs for at least two years. This information have been added in L151 to 154.